# Metabolic Abnormalities in Glioblastoma and Metabolic Strategies to Overcome Treatment Resistance

**DOI:** 10.3390/cancers11091231

**Published:** 2019-08-23

**Authors:** Weihua Zhou, Daniel R. Wahl

**Affiliations:** Department of Radiation Oncology, University of Michigan, Ann Arbor, MI 48109, USA

**Keywords:** glioma, glioblastoma, metabolism, metabolic remodeling, metabolic targeting, radiation

## Abstract

Glioblastoma (GBM) is the most common and aggressive primary brain tumor and is nearly universally fatal. Targeted therapy and immunotherapy have had limited success in GBM, leaving surgery, alkylating chemotherapy and ionizing radiation as the standards of care. Like most cancers, GBMs rewire metabolism to fuel survival, proliferation, and invasion. Emerging evidence suggests that this metabolic reprogramming also mediates resistance to the standard-of-care therapies used to treat GBM. In this review, we discuss the noteworthy metabolic features of GBM, the key pathways that reshape tumor metabolism, and how inhibiting abnormal metabolism may be able to overcome the inherent resistance of GBM to radiation and chemotherapy.

## 1. Introduction

Glioblastoma (GBM) is the most prevalent and lethal primary brain tumors in adults, with a median overall survival of approximately 15 months [1,2,3]. Standard therapy for GBM consists of maximal surgical resection, followed by radiotherapy and concurrent and adjuvant temozolomide (TMZ; an orally available alkylating chemotherapy) [4,5]. Despite this multimodality treatment, GBMs inevitably recur. Recurrences are typically local (i.e., within the high dose radiation field) and extracranial metastases are extraordinarily rare [6].

Therapeutic options are limited following recurrence and include re-resection, re-irradiation, additional cytotoxic chemotherapy, and bevacizumab, a humanized vascular endothelial growth factor monoclonal antibody that inhibits angiogenesis [7]. Upon recurrence, the average survival time for a patient with GBM is approximately six months, which reemphasizes the need to develop novel therapeutic approaches [8].

The search for new therapies for GBM has largely centered on the molecular targets uncovered through large-scale genomic analysis efforts such as The Cancer Genome Atlas. Most GBMs harbor recurrent molecular alterations disrupting core pathways involved in regulation of growth (*EGFR* [9,10], *PDGFRA* [11], *FGFR* [12,13], mitogen-activated protein kinase (MAPK) [14,15] and phosphoinositide 3-kinase (PI3K) [16,17] signaling pathways), cell cycle [18,19,20,21], autophagy [22], DNA repair, and apoptosis [23,24], as well as regulators of angiogenesis [25,26] and immune checkpoints [27,28,29]. Unfortunately, personalized therapies targeting these genomic alterations have not yet been successful in the clinic, possibly due to the intense intratumoral heterogeneity that characterizes GBM [30]. New treatment strategies that may be efficacious despite intratumoral genomic heterogeneity are urgently needed.

Altered cellular metabolism is a hallmark of cancer. Like most cancers, GBMs rewire metabolism for numerous pro-growth and -survival functions including macromolecule synthesis, ATP generation and antioxidant regeneration. Distinct oncogenic alterations (e.g., *MYC* amplification, *PTEN* deletion, and *TP53* mutation) can activate common metabolic adaptations such as increased glycolytic activity and lead to tumor cell growth or progression [31]. Since many of the metabolic adaptations of GBM can exist across numerous genotypes, it is possible that a therapeutic strategy of targeting abnormal metabolism may meet more success than targeting genomic alterations in a heterogeneous tumor such as GBM. Hence, GBM metabolism is an area of intense research to identify novel therapeutic targets and biomarkers [32]. In this review, we firstly review the updated molecular classification of glioma and GBM, discuss the metabolic features of GBMs, and highlight how specific metabolic processes regulate tumorigenesis and progression in GBM. Finally, we will discuss how targeting abnormal GBM metabolisms may overcome resistance to standard therapies.

## 2. Histopathologic and Molecular Classification of Diffuse Gliomas

Diffuse gliomas are the most frequent primary adult malignant brain tumors and have historically been diagnosed based on their histologic resemblance to astrocytes or oligodendrocytes using conventional microscopy [33,34]. In 2016, the World Health Organization (WHO) revised their classification system of diffuse gliomas to account for definitional molecular alterations including mutations in isocitrate dehydrogenase (IDH) and whole-arm deletions of the chromosomal arms 1p and 19 q (termed 1p/19q codeletion) [34,35] (Figure 1). Oligodendrogliomas are now defined as having both mutations of IDH and codeletion of 1p/19q. Oligodendrogliomas are low grade (WHO grade 2) if they have little morphologic atypia under conventional microscopy and anaplastic (WHO grade 3) if they have aggressive morphologic features such as increased mitotic activity, microvascular proliferation, or necrosis. When treated with combined radiation and chemotherapy, patients with grade 2 or 3 oligodendrogliomas have favorable survival times (median >10 years) [36,37]. IDH-mutant astrocytomas are defined by the presence of an IDH mutation and the absence of 1p/19q codeletion. These tumors also frequently harbor mutations in p53 and ATRX, though these alterations are not definitional. Patients with IDH mutant astrocytomas that are grade 2 (termed low grade or diffuse) or grade 3 (termed anaplastic, based on the presence of increased mitotic activity) and receiving optimal therapy have a median overall survival on the order of 8–12 years depending on the study [38]. Grade 2 and grade 3 IDH-mutant astrocytomas often develop more aggressive behavior, which is accompanied by microscopic features such as necrosis and microvascular proliferation and additional molecular changes such as CDKN2A/B loss. These changes are characteristic of an IDH mutant GBM (IDHmutGBM), which are often termed “secondary” GBM due to their transformation from lower grade tumors. Patients with IDHmutGBM have median survival times between two and three years [39,40]. Diffuse astrocytomas without IDH mutations are aggressive tumors that often contain EGFR amplification, TERT promoter mutation, and/or aneuploidy in chromosomes 7 and 10. When IDH-wild type diffuse gliomas have aggressive histologic features such as necrosis and microvascular proliferation they are termed GBM (WHO grade 4). These IDHwtGBM account for more than 90% of all GBM and, unlike IDHmutGBM, arise de novo and have short natural histories. The median survival time for patients with IDHwtGBM is on the order of 1–1.5 years [4,39]. Patients with diffuse astrocytomas that lack the IDH mutation but whose tumors lack the histologic criteria to be called GBM have natural histories that more closely resemble IDHwtGBM than their grade 2 or 3 IDH-mutant counterparts [41]. Like their natural history and etiology, the metabolic characteristics and liabilities of GBM with and without IDH mutations are distinct and described separately below. A recently described discrete type of high grade glioma (the H3K27 mutant diffuse midline glioma) is definitionally grade 4 based on its molecular rather than histologic features, but its associated metabolic abnormalities fall outside the scope of this review [42].

## 3. Metabolic Characteristics of IDHwtGBM

### 3.1. Glycolysis

Glycolysis is responsible for catabolizing glucose to pyruvate, which can be converted to lactate and secreted or enter the TCA cycle via either pyruvate dehydrogenase (to generate NADH and ATP) or pyruvate carboxylase (to contribute to anaplerosis, Figure 2). GBM is associated with a significant increase in glycolysis for energy production [43]. Hexokinase (HK) catalyzes the first glycolytic step to generate glucose-6-phosphate, with the isoform HK2 strongly expressing in cancers including GBM [44]. *ENO1*, an isoform of the glycolytic gene enolase generating phosphoenolpyruvate (PEP), is deleted in GBM, and *ENO2* silencing selectively inhibits growth, survival and tumorigenic potential of *ENO1*-deleted GBM cells [45]. Knockdown of glycolytic genes (*HK2*, *PFKP*, *ALDOA*, *PGAM1*, *ENO1*, *ENO2,* and *PDK1*) strongly inhibits GBM growth [46], indicating that key glycolytic enzymes are essential for GBM growth.

The metabolic proteins of glycolysis in IDHwtGBM are frequently regulated by oncogenic signaling pathways. Ras, serving as an upstream pathway, inhibits pyruvate dehydrogenase (PDH) activity through downregulation of PDH phosphatase (PDP) expression and promotes GBM tumorigenesis [47]. AKT binds to and phosphorylates the PFK1 platelet (PFKP) isoform and elevates PFKP expression in GBM, leading to cell proliferation, tumor growth, and poor prognosis [48]. Hypoxia, EGFR, K-Ras-G12V, and B-Raf-V600E can all stimulate glycolysis by inducing the translocation of PGK1 to the nucleus, which phosphorylates and activates PDHK1. This signaling inhibits PDH activity and promotes GBM tumor development [49]. High-level amplification of *MYC* and *MYCN* genes are also observed in GBM [11,50,51], and inhibition of Myc-induced glycolysis selectively killed *MYC/MYCN*-amplified patient-derived GBM tumor spheres and extended the survival of mice bearing *MYCN*-amplified PDX [49]. The mechanistic target of rapamycin complex 1 (mTORC1) and 2 (mTORC2) are frequently aberrantly activated in GBM. Intriguingly, mTORC2 regulates glycolytic metabolism in GBM through Akt-independent phosphorylation of class IIa histone deacetylases (HDACs) to release c-Myc [52].

Pyruvate kinase (PK) catalyzes the final step in glycolysis by irreversibly transferring the phosphate group from phophoenolpyruvate (PEP) to produce pyruvate and ATP. PK consists of four isoforms encoded by two genes, *PKLR* and *PKM*. Alternative splicing of *PKM* pre-mRNA further generates PKM1 and PKM2 isoforms [53]. PKM2, but not PKM1, is selectively expressed in proliferating or tumor cells [54,55], and can disassociate HDAC3 from the *MYC* promoter to promote GBM tumorigenesis through directly binding and phosphorylating histone H3 [56]. EGFR activation in GBM induces translocation of PKM2 into the nucleus to transactivate β-catenin, which mediates the upregulation of c-Myc-dependent glycolysis and cell proliferation [57]. During oxidative stress, PKM2 can translocate to the mitochondria and phosphorylate Bcl2 to inhibit apoptosis directly, promoting GBM cell growth and correlating with poor prognosis [58]. However, the ability of PKM2 to act as a kinase has recently come into question [59], which makes it possible that its oncogenic properties may be due to its metabolic function. Glycolysis can also be regulated by macrophages, which enhanced PDPK1-mediated phosphorylation of PGK1 by secreting interleukin-6 (IL-6), leading to enhanced glycolysis and GBM tumorigenesis [60].

The high utilization of glucose to support GBM growth and survival indicates that inhibiting glycolysis in GBM may have a therapeutic value. Depletion of HK in intracranial xenograft models of IDHwtGBM decreased tumor proliferation and angiogenesis, but increased invasion [61], and the latter may impact the therapeutic suitability of targeting this pathway. Tumors expressing HK2 may also express HK1, rendering them resistant to HK2 inhibition. Moreover, other metabolically active tissues, including skeletal muscle, cardiac, and adipose tissues, also express HK2 [62], indicating that the anti-tumor effects of HK2 inhibition may be challenged or limited by non-specificity and systemic toxicity. However, systemic knockout of *HK2* in mice bearing lung cancer has shown profound therapeutic effects on tumor metabolism and growth, with minimal adverse effects to normal mouse physiology [63]. Additionally, the glucose analog, 2-deoxyglucose (2DG), an inhibitor of the glycolytic pathway, has been tolerated reasonably well in early clinical trials for patients with cancer [64,65]. Thus, despite the dependence of normal brain on glucose, there may be a therapeutic window for targeting glycolytic metabolism in GBM [66].

### 3.2. TCA Cycle and Acetate

The tricarboxylic acid (TCA) cycle, also known as citric acid or the Krebs cycle, is a central metabolic hub that provides energetic intermediates and/or anabolic precursors through oxidization of acetyl-CoA into carbon dioxide. Both activated oncogenes and deleted tumor suppressors dysregulate the TCA cycle across cancers, making it a possible therapeutic target [67]. 

In GBM, the major carbon sources for the TCA cycle appear to be glucose and acetate. This mirrors the normal brain, where glucose has long been considered the dominant TCA carbon source, but where other metabolites, such as fatty acids (octanoate), acetate, and ketones, can be used as alternate fuels, especially during hypoglycemia [68,69]. These alternative energy fuels can also drive GBM growth and survival. Indeed, less than 50% of the GBM acetyl-CoA pool is derived from glucose in vivo [70]. Further study in intracranial GBM xenografts showed that acetate contributes a significant fraction of carbon to the TCA cycle in GBMs through the action of acetyl-CoA synthetase enzyme 2 [71]. In contrast to glucose and acetate, circulating glutamine appears to contribute little carbon to the GBM TCA cycle. Rather, many GBMs express glutamine synthetase and utilize astrocyte-derived glutamate to generate the glutamine needed for cellular function [71,72].

Recently, lactate has been reported as an important TCA cycle carbon source in normal tissues and cancers such as non-small cell lung cancer (NSCLC) [73]. The brain appears to behave differently and incorporates little circulating lactate-derived carbon into the TCA cycle [74]. Whether GBM utilizes circulating lactate as a fuel source, or if a separate intracranial pool of lactate exists that can fuel the tumor or normal brain are unanswered questions.

Isocitrate dehydrogenases (IDHs) are a group of enzymes playing a crucial role in the TCA cycle by catalyzing the oxidative decarboxylation of isocitrate to α-ketoglutarate (α-KG), using nicotinamide adenine dinucleotide phosphate (NADP^+^) or nicotinamide adenine dinucleotide (NAD^+^) as a cofactor to generate NADPH or NADH during catalysis (Figure 2). Three different IDH paralogs (IDH1, IDH2, and IDH3) have been identified, with IDH1 performing its function in the cytosol and peroxisomes, whereas IDH2 and IDH3 function in the mitochondria [75,76]. IDH1 and IDH2 play important roles in many cellular functions, including glucose sensing [77], lipogenesis [78,79], glutamine catabolism [78], and cellular defense against reactive oxygen species and radiation [80,81]. 

IDHwtGBMs upregulate wtIDH1, which fuels tumor growth and therapy resistance. Knockdown of wtIDH1 in IDHwtGBM cell line models depletes NADPH, deoxynucleotides, and antioxidants and makes these models more sensitive to radiotherapy [82]. A complementary study confirmed that wtIDH1 plays an important role in maintaining IDHwtGBM antioxidant levels and drives lipid biosynthesis in these models. This study showed that pharmacologic or genetic inhibition of wtIDH1 increased the sensitivity of these model systems to targeted therapy [83]. Together, these studies suggest that the therapeutic targeting of wtIDH1 may be a promising therapeutic strategy in IDHwtGBM.

### 3.3. Glutamine Metabolism

Glutamine circulates at high concentrations (~500 μM) and is a carbon and nitrogen source to support biosynthesis, energetics, and cellular homeostasis for cancer cell growth [84]. The total concentration of glutamine is greater in GBM tissue than the surrounding normal brain [43] and the ability to metabolize this glutamine is critical for GBM proliferation and survival [85]. Somewhat surprisingly, circulating glutamine appears to be a minor contributor to the intracranial glutamine pool [86]. Rather than coming from the circulation, brain glutamine is primarily synthesized from glutamate and ammonia in situ by glial cells expressing glutamine synthetase (GS) (Figure 2) [87]. GS expression and function appears important for GBM growth and survival. Patients who GBMs express low or absent levels of GS have approximately two-fold longer survival compared to patients whose tumors express high levels of GS [88]. GS expression is further elevated in the stem-like cells thought to drive GBM recurrence and treatment resistance [89]. The glutamine generated from astrocytes or GS-positive GBM cells supports tumor nucleotide biosynthesis and growth, suggesting that its inhibition may have therapeutic effects.

Glutamine breakdown may also be important for GBM growth and survival. Glutaminase (GLS), which converts glutamine to an ammonium ion and glutamate (Figure 2), is encoded by kidney-type glutaminase (GLS1) and liver-type glutaminase (GLS2) [90]. GLS has been proposed as a therapeutic target in many cancers and clinical trials of glutaminase inhibitors are ongoing, including for patients with GBM (NCT03875313). Targeting GLS slows the growth of subsets of GBMs in vitro and in vivo [91]. The molecular underpinnings of GLS dependence in GBM are under investigation. Models resembling the mesenchymal GBM subtype may be especially sensitive to GLS inhibition [92] as are GBMs that express mutant IDH [93] or that have hyperactive c-Myc [94]. GLS may also play a role in mediating GBM therapy resistance. Elevated GLS and glutamate levels following mTOR inhibition are responsible for the resistance of GBM cells to this compound, and dual inhibition of mTOR and GLS synergistically slows GBM growth in vivo [85].

Together, these studies suggest that targeting glutamine metabolism may have a therapeutic utility for patients with GBM. Particular molecular features such as c-Myc amplification or the mesenchymal transcriptional subtype may be most benefited by this strategy. Patients whose tumors carry the IDH mutation are also excellent candidates for inhibition of glutamine metabolism, and this concept is discussed in more detail in the IDH mutations subsection. Since cortical glutamine levels are generated in situ, it appears that small molecule inhibitors of glutaminase or glutamine uptake may be more effective than dietary modifications to alter circulating glutamine levels [86,89].

### 3.4. Lipid Metabolism

Lipids are a class of fat-soluble organic compounds that have important structural, signaling and energetic functions in many cancers including GBM. Common lipid types include fatty acids, triglycerides (a glycerol molecule annealed to three fatty acids), phospholipids (a glycerol molecule annealed to two fatty acids and one polar phosphate-containing head group), and cholesterol. GBMs exist in a lipid-rich environment, as the dry weight of normal human brain contains more than 50% lipid and the brain itself contains about a quarter of the body’s cholesterol [95]. Since much of this lipid is housed in structural elements of normal neural tissue such as myelin, the amount of environmental lipid available to a GBM may be relatively low. Indeed, increasing evidence suggests that GBMs must synthesize significant amounts of lipids to generate the necessary membrane components and lipid signaling molecules needed to proliferate [96].

To synthesize fatty acids (the main component of most lipids) de novo, GBMs must first generate cytosolic acetyl-CoA, which can be formed from citrate via the action of ATP citrate lyase, or from acetate via the action of acetyl-CoA synthetase (ACSS; Figure 2) [97]. The generation of cytosolic acetyl-CoA in GBM is regulated by both oncogenes and tumor suppressors. Indeed, the concomitant introduction of a *BRAF*^V600E^ mutation and deletion of *TP53* and *PTEN* in astrocytes increased by seven-fold the expression of ACSS2, which contributes to the cytosolic pool of acetyl-CoA, but had little effect on ACSS1, which generates mitochondrial acetyl-CoA [71]. This same paper showed that knockdown of ACSS2 decreased the GBM neurosphere growth and viability, and the authors postulated that this phenotype is due to decreased oxidation of acetyl-CoA in the mitochondria (as discussed above). However, it is possible that ACSS2 promotes GBM growth by funneling extracellular acetate towards lipogenesis in addition to oxidation in the TCA cycle.

De novo lipogenesis continues when cytosolic acetyl-CoA is carboxylated by the enzyme acetyl-CoA carboxylase (ACC) in an ATP-dependent fashion to generate malonyl-CoA. The ACC reaction is the rate limiting step in de novo lipogenesis and it is followed by the reactions of fatty acid synthase (FASN), which catalyzes the multi-step synthesis of the fatty acid palmitate from seven molecules of malonyl-CoA, a single molecule of acetyl-CoA, seven molecules of ATP and 14 molecules of NADPH. The activity of de novo lipogenesis is important for GBM growth. In cell culture models of GBM, forced expression of the vIII variant of EGFR stimulates de novo lipogenesis and growth, which is reversed when ACC is inhibited with small interfering RNAs (siRNAs) [98]. Pharmacologic inhibition of de novo lipogenesis using the adenosine monophosphate-dependent protein kinase (AMPK)agonist 5-Aminoimidazole-4-carboxamide ribonucleotide (AICAR) causes a similar anti-proliferative effect in EGFR-driven GBM models both in vitro and in vivo [99]. Pharmacologic and genetic inhibition of FASN similarly inhibits the proliferation of GSC cell lines [100] and immortalized GBM cell lines [101]. Clinical grade inhibitors of FASN have been developed and are being investigated in numerous cancers including GBM (NCT03032484). 

The dependence of GBM on de novo lipogenesis is governed by increased oncogenic activity, which in turn stimulates sterol regulatory element-binding protein-1 (SREBP-1), a transcription factor, and master regulator of lipogenesis [102,103]. EGFR promotes N-glycosylation of SREBP cleavage-activating protein (SCAP) to active SREBP and drive GBM growth [104,105]. Inhibition of sterol O-acyltransferase (SOAT1) blocks cholesterol esterification and also suppresses GBM growth through blocking SREBP-1-regulated fatty acid synthesis [106]. SREBP is activated by other GBM drivers including PI3K/mTORC1 [107]. Due to the genomic heterogeneity characteristic of GBM, it is possible that the inhibition of SREBP or de novo lipogenesis itself may be a fruitful and unexplored therapeutic strategy in GBM.

In some contexts, the oxidation of fatty acids, rather than their synthesis, seems to govern GBM growth and survival. There is increasing evidence that a population of slowly cycling stem-like cells drives GBM growth and recurrence after therapy [108]. Such “slowly cycling” GBM progenitor cells may depend more heavily on fatty acid oxidation than synthesis [109]. Indeed, fatty acid binding protein 7 (FABP7), a lipid chaperone that mediates fatty acid uptake and subsequent oxidation, is highly expressed in GBM neurospheres and slowly growing progenitor cells, and its inhibition slows the growth and invasiveness of GBM models in vitro and in vivo [109,110]. In support of this hypothesis, primary cultured human GBM cells utilize fatty acids as a primary component of their oxidative metabolism and inhibition of fatty acid oxidation with the compound etomoxir slowed the proliferation of primary GBM cultures in vitro and their tumorigenesis in intracranial mouse models [111]. These findings suggest that lipid metabolism, like most other phenotypes in GBM, is heterogeneous and that combination therapies may be needed to improve patient outcomes. One potential strategy would be to inhibit both fatty acid synthesis (to affect the rapidly dividing GBM progenitor cells) while simultaneously inhibiting fatty acid oxidation (to affect slowly cycling stem-like cells). However, the therapeutic window for such combined inhibition of oxidation and synthesis is unlikely to be favorable. Since slow cycling cells that depend on fatty acid oxidation may drive recurrences after chemotherapy and/or radiation, a more promising strategy may be to combine etomoxir with initial chemoradiation in patients with GBM.

Like fatty acids, cholesterol can be synthesized de novo or taken up from the environment. Both pathways are important for GBM growth and survival. EGFR-mediated SREBP1 activation in GBM promotes the expression of the low density lipoprotein receptor (LDLR), which is a key mediator of cholesterol uptake [102]. Agents that force the degradation of LDLR and block cholesterol uptake slow the growth of GBM cells in vitro and tumors in vivo. Systemic inhibition of LDLR may have a limited therapeutic window as genetic deletion of this target causes severe hypercholesterolemia and pharmacologic modulation appears to cause neuropsychiatric side effects [112,113]. The de novo synthesis of cholesterol is governed by the activity of hydroxylmethylglutaryl-CoA reductase (HMG-CoA reductase), which is inhibited by the statin drug class. There is conflicting epidemiologic evidence regarding whether the use of statins is associated with improved survival in patients with GBM [114]. Drugs in the statin class have anti-proliferative effects when used alone or in combination with DNA damaging agents in patient-derived GBM cell lines, but the mechanism accounting for this efficacy is uncertain [115]. Similar combination therapies are now being tested in the clinic in a small phase 2 trial, but final results have not yet been published [113].

### 3.5. Nucleotide Metabolism

Nucleotides are important signaling and structural molecules and are the principal carriers of accessible chemical energy in the cell. Nucleotides consist of a nitrogenous base attached to a sugar molecule (ribose or deoxyribose), which is coupled to phosphate groups. Structurally, nucleotides and deoxynucleotides are principal components of DNA, RNA, and ribosomes. Production of nucleotides is carried out by two main pathways, termed de novo nucleotide synthesis and nucleotide salvage. 

De novo synthesis of purines or pyrimidines utilizes one carbon unit, amino acids, ribose, and significant amounts of free energy to generate new nitrogenous base rings from scratch. By contrast, nucleotide salvage involves the conversion of pre-formed bases into nucleotides and requires less energy than de novo synthesis. Conventionally, proliferating cells have a higher reliance on de novo nucleotide synthesis than nucleotide salvage pathways and this appears to be the case in GBM as well. De novo purine synthesis is highly elevated in GBM stem cells and promotes cell growth and tumor formation, which is mediated and maintained by the activity of c-MYC [116]. In some neurosphere models of GBM, de novo pyrimidine synthesis also plays an important role [117]. Salvage pathways are also active in GBM. Nuclear medicine scans using the radiolabeled tracer ^18^F-fluorothymidine routinely show an increased signal in GBM tissue compared to nearby normal brain, but it is not entirely clear whether this finding is due to increased pyrimidine salvage in GBM or simply disruption of the blood brain barrier [118]. Purine salvage pathways are also intact in GBM. Indeed, the ability to scavenge hypoxanthine, the most abundant purine in the CSF, may account for the resistance of GBM to anti-folate therapy [119]. Inhibitors of de novo purine synthesis that act downstream of the hypoxanthine salvage step (which generates inosine monophosphate) may be more effective. There have been preliminary clinical investigations of inhibitors of nucleotide metabolism in GBM. Gemcitabine, a cytidine analog that can be incorporated into DNA or slow deoxynucleotide production by inhibiting ribonucleotide reductase, can cross the blood brain barrier and accumulate into tumors such as GBM at active concentrations [120,121]. Consistent with these observations, a clinical trial combining gemcitabine and radiation for patients with high-grade glioma including GBM showed reasonable safety and promising clinical outcomes [122]. Since nucleotide metabolism is a tractable and established therapeutic target, we are optimistic that the increased understanding of altered nucleotide metabolism in GBM will result in therapeutic advances.

### 3.6. Alternative Metabolic Pathways and Plasticity

In addition to dysregulation of these canonical metabolic pathways, many other metabolic liabilities in GBM are emerging. The importance of one-carbon reactions across multiple cancers also holds true in GBM. Overexpression of mitochondrial serine hydroxymethyltransferase (SHMT2) and glycine decarboxylase (GLDC) in GBM promotes tumor growth by inhibition of PKM2 activity and reduction of oxygen consumption [123]. 5-methylthioadenosine phosphorylase (MTAP), a key enzyme in the methionine salvage pathway, is deleted in almost half of GBM tumors [124]. MTAP deletion increases dependence on protein arginine methyltransferase 5 (PRMT5) and inhibition of PRMT5 selectively kills MTAP-null cancer cells [125,126,127], indicating that inhibition of PRMT5 is a potential therapy for MTAP-deleted tumors. Further in vitro and in vivo studies have demonstrated the antitumor effects of PRMT5 inhibitors in GBM and underscored the importance of developing it in the clinic [128]. Clinical trials are underway (NCT02783300) to evaluate the escalated dosage of a PRMT5 inhibitor in a variety of solid tumors including GBM. The clinical development of inhibitors of one-carbon metabolism has been more fraught and clinical trials are not currently ongoing [129]. A more tractable strategy may be to alter one-carbon metabolism by altering diet. Indeed, humans and mice on a methionine-restricted diet had depleted circulating antioxidant and nucleotide levels and this dietary modification increased the responsiveness of tumors in mice to both radiation and chemotherapy [130].

## 4. IDH1/2 Mutation and 2-Hydroxyglutarate

In 2008, exome-sequencing studies identified a novel mutation of *IDH1* in 12% of GBM patients [131]. Further studies have found that *IDH1*, or *IDH2* mutation, happens in ~80% of WHO grade II–III gliomas and secondary GBM [40]. Mutations in *IDH1* mainly affect R132, which is the binding site for isocitrate, and R132H is the most common alteration, comprising >80% of all *IDH1* mutations in gliomas [40,131,132,133]. Mutations in *IDH2* exclusively affect R172 and R140, with the former being structurally analogous to *IDH1* R132 [40,132]. In addition to reducing affinity for isocitrate and losing its normal catalytic activity [134], mutant *IDH1* or *IDH2* also gained the function of catalyzing the reduction of α-KG to produce the (R) enantiomer of 2-hydroxyglutarate, (R)-2HG (also known as D-2HG), which accumulates in IDH1 or IDH2 mutated gliomas to millimolar concentrations [135] (Figure 2). Most IDH1 mutant tumors still retain one wild-type *IDH1* allele and disruption of the residual wild-type *IDH1* allele decreases D-2HG production, indicating that IDH1 mutant-induced D-2HG production is probably dependent on wild-type IDH1 [136]. Initial studies suggested that this dependence might be due to the substrate channeling or cooperative effects exiting in the IDH1 wt/mut heterodimers [137,138], however the recently published crystal structure of the IDH1 wt/mut heterodimer fails to show physical association between the active sites of the wild-type and the mutant, which suggests that substrate channeling does not occur [139].

A complete understanding of how mutant IDH and D-2HG affects GBM cells is still developing. The *IDH1* mutation contributes to tumorigenesis through regulation of the HIF-1α pathway, alteration of histone and DNA methylation, activation of glutaminolysis, and increased sensitivity to glucose deprivation [134,140,141] (Figure 2). These tumorigenic properties of mutant IDH are due to its product (D)-2HG, which has accordingly been considered an oncometabolite. Due to its structural similarity to alpha-ketoglutarate (α-KG), D-2HG inhibits DNA and histone demethylases, namely the ten-eleven translocation enzymes (TETs) and lysine demethylases (KDMs) [142,143], to block differentiation of non-transformed cells [140]. How the IDH1 mutation affects HIF-1α activity is still controversial. IDH1 mutant glioma cells reduce the production of α-KG and increase HIF-1α and its target genes (*GLUT1, VEGF*, and *PDK1*) to stimulate tumor growth and angiogenesis [134], whereas IDH1 mutants have also been shown to stimulate EglN prolyl 4-hydroxylase, which destabilizes HIF-1α and diminishes the expression of its target genes to promote transformation [144]. Mutant IDH1 cooperated with PDGFA and inactivation of *CDKN2A, ATRX*, and *PTEN* to promote glioma development [145]. Moreover, pyruvate dehydrogenase (PDH) activity is downregulated in a 2HG-dependent manner in IDH mutant GBM cells, leading to reduced pyruvate flux into the TCA cycle and decreased glutamate levels, and finally promoted colony formation and cell proliferation [141].

Targeting mutant IDH to normalize (D)-2HG levels is an attractive cancer therapeutic strategy in gliomas. Inhibitor of mutant IDH1 (AGI-5198) impaired the growth of IDH1-mutant but not IDH1-wild-type glioma cells by increasing the expression of genes associated with gliogenic differentiation [146]. Phase I studies of another two mutant IDH1 inhibitors (AGI-881, NCT02481154; AG-120) showed a favorable safety profile and potential efficacy in patients with IDH mutated gliomas [147]. Promisingly, ivosidenib (AG-120) has been approved by FDA as first-line treatment for acute myeloid leukemia (AML) with *IDH1* mutation after a positive clinical trial (NCT02074839), further indicating the therapeutic potential of targeting this alteration. 

While much data suggests that inhibition of D-2HG production may benefit patients with IDHmutGBM, there are several lines of evidence that suggests an alternative approach of targeting not D-2HG itself, but rather the vulnerabilities that bestows on IDH mutant tumors. Patients with IDHmutGBM have a better prognosis than those with IDHwtGBM [131], but it is not clear if this is due to D-2HG-mediated vulnerabilities or different natural histories between these two tumor types. A recent study showed that inhibition of Bcl-xL, only in the presence of D-2HG, could significantly induce more apoptosis in IDH1-mutant cells than wild-type IDH1 cells [148]. The reason is that D-2HG-mediated energy depletion activates AMPK and then blunts protein synthesis and mTOR signaling, leading to a decline of Mcl-1, which sensitizes glioblastoma cells to Bcl-xL inhibition-mediated apoptosis. Furthermore, mutant IDH1 decreases the HIF-1α-responsive gene, *LDHA,* which is essential for glycolysis and is overexpressed in cancers, through IDH-mutant-induced methylation of *LDHA* promoter [149], finally limiting the rapid cell growth of high-grade glioma. D-2HG also inhibits ATP synthase and mTOR signaling downstream to decrease tumor cell growth and viability [150]. D-2HG also inhibits the branched-chain amino acid transaminase 1 (BCAT1) and 2 (BCAT2), which renders IDH1 mutant glioma cells dependent on glutaminase and increases their sensitivity to oxidative stress [151].

There is controversy as to whether the IDH mutation promotes or impairs the ability of gliomas to repair DNA damage. D-2HG has been reported to inhibit homologous recombination by inhibiting the activity of histone lysine demethylases, thereby increasing the sensitivity of these cancers to inhibitors and alkylating agents such as temozolomide [145,152,153]. Consistent with these findings, inhibition of 2HG production makes IDH mutant cell line models of glioma resistant to radiation [154]. However, a genetically engineered IDH mutant astrocytoma, in which p53 and ATRX were also inactivated, showed an enhanced ability to repair DNA due to upregulation of homologous recombination [155]. In this model, the IDH mutation increased the expression of key components of the DNA damage response by increasing histone methylation. These epigenetic changes resulted in profoundly radioresistant tumors, and this resistance could be overcome with inhibitors of ATM or CHK1/2. These apparently discordant results have important complications about the best strategies to design combination treatments for patients with IDHmutGBM and a final answer will likely have to wait for the clinical trials themselves. 

Thus, the effects of the IDH1 mutation and D-2HG in GBMs may be context-dependent, which has important implications for therapy decisions. Early in tumorigenesis, it may be beneficial to inhibit the production of D-2HG to limit the initial transforming events. However, once a tumor is established, it may become less dependent on D-2HG for growth and that inhibition of mutant IDH may be less effective [156]. In this situation, a more promising strategy may be to inhibit the D-2HG-induced therapeutic vulnerabilities. This approach is supported by preclinical evidence. In one study, D-2HG depletion failed to inhibit the growth of established IDH1 mutant glioma cell lines and tumors, but rendered these tumors exquisitely sensitive to NAMPT inhibition [157]. This phenotype occurred because D-2HG inhibits the NAD^+^ producing enzyme Naprt1, which causes IDH mutant cells to preferentially rely on NAMPT for NAD^+^ generation. Thus, while inhibitors of mutant IDH may prevent tumor formation or growth early in the disease course, their ability to reduce D-2HG may reverse some of the therapeutically attractive vulnerabilities of IDH mutant tumors. 

## 5. Metabolic Targeting to Sensitize GBM to Standard Therapies

### 5.1. Radiation

Radiotherapy is one of only a handful of treatments that improves survival in patients with GBM, and the modulation of radiation resistance is of significant interest to further improving outcomes. Aberrant metabolic pathways intersect with many mediators of radiation resistance and targeting these abnormalities may be helpful to improve GBM radio-sensitivity (Figure 3).

High rates of glycolysis correlate with radiation resistance in multiple cancer models and inhibition of glycolysis can help overcome this resistance. Knockdown of the glycolytic enzyme HK2, reduced GBM growth in vitro and in vivo by increasing radiation-induced apoptosis [61,158]. Pharmacologic inhibition of upper glycolysis with 2-DG shows radio-sensitizing effects in multiple solid tumors, including GBM [65]. Inhibiting lower glycolysis has similar effects. Dichloroacetate (DCA) activates pyruvate dehydrogenase by inhibiting its regulatory kinase (pyruvate dehydrogenase kinase). Thus DCA promotes the oxidation of glucose and limits its conversion to lactate [159,160]. DCA can augment the DNA double-strand breaks induced by radiotherapy and effectively sensitize GBM to radiotherapy in both in vitro and in vivo models by reversing radiotherapy-induced glycolytic metabolism under hypoxia conditions [161]. DCA crosses the blood–brain barrier and is well tolerated in patients, but its efficacy in GBM is uncertain [162]. Whether DCA retains its radiosensitizing properties in patients with GBM is unknown.

NADPH is a critical reductant that facilitates survival in the face of numerous pro-oxidants, including radiation. IDH1 is the most upregulated NADPH-producing enzyme in IDHwtGBM [83] and knockdown of wild-type *IDH1* reduced NADPH levels and radiosensitized GBM in vitro and in vivo by inducing accelerated cellular senescence [82]. These data suggest that inhibiting wild type IDH in IDHwtGBM might be a reasonable therapeutic strategy. 

Whether mutant IDH should be inhibited in combination with radiation is less straightforward. As discussed above, the IDH1 mutation causes the accumulation of 2HG, which decreases the ability of gliomas to repair DNA [153]. These data suggest that mutant IDH tumors may be more sensitive to radiation and that inhibition of 2HG may be radioprotective. Indeed, although inhibition of IDH1 mutant with a selective IDH1mt inhibitor (AGI-5198) impaired glioma tumor growth in vitro and in vivo by blocking the production of 2HG [146], this IDH1mt inhibitor was reported to radio-protect IDH1 mutant cancer cells by decreasing radiation-induced ROS levels, DNA double-strand breaks and cell death, indicating that IDH1mt inhibition limits radiation efficacy [154]. If, on the other hand, 2HG actually promotes DNA repair by promoting ATM expression and HR activity, it may make sense to combine radiation and inhibition of mutant IDH [155]. Further study is needed to determine whether or how best to combine radiotherapy with inhibition of mutant IDH.

NAD^+^ is a critical metabolic co-factor that also plays a role in the DNA damage response. Poly(ADP-ribose) polymerases (PARPs), key enzymes in the base excision repair (BER) and single-strand break repair (SSBR) pathways [163], are major consumers of NAD^+^ [164]. The catalytic activity of PARP-1 converts NAD^+^ into long and branched chains of PAR, which rapidly recruit DNA repair factors to compromised sites of DNA damage [165]. PARP inhibitors, which compete with NAD^+^ at the catalytic site of PARPs, or trap PARPs to DNA, increase the radio-sensitivity of GBM in both in vitro and in vivo studies [166,167,168]. Other strategies to limit NAD^+^ availability have similar effects on radiosensitivity. Nicotinamide phosphoribosyltransferase (NAMPT) is the rate-limiting enzyme in the NAD^+^ salvage pathway that converts nicotinamide to NAD^+^. NAMPT is up-regulated in GBM stem-like cells following radiation and knock down or pharmacological inhibition of NAMPT reversed radio-resistance [169]. Clinical trials of NAMPT inhibitors have been performed as monotherapy, but were discontinued due to toxicity and minimal activity [170]. Whether these agents would have had efficacy in combination with radiation is unknown. 

When oxygen is present, conventional radiotherapy mediates the majority of its DNA damage through reactive oxygen species (ROS) such as O_2_^−^, H_2_O_2_, and –OH. Thus, potentiating ROS levels or depleting antioxidants are rationale metabolic strategies to augment the radiation response. High dose ascorbate acid (AA) dramatically increases ROS within cancer cells, either due to its ability to inhibit glycolysis or through interactions with labile iron [171,172,173]. Furthermore, high dose ascorbate induces double stranded breaks (DSBs) and radio-sensitizes GBM cells through a ROS-mediated mechanism [174]. This combination strategy is now being explored clinically in a variety of cancers including glioma with promising early results (NCT02344355) [175,176]. 

Glutamine metabolism is also involved in modulating ROS and oxidative stress, because its metabolism by GLS generates glutamate, one of the three amino acids used to synthesize the key antioxidant glutathione [177,178]. IDHmtGBMs are exquisitely dependent on GLS to generate glutamate, because their high levels of (R)2HG inhibit the alternative BCAT pathway of glutamate generation. Thus, treatment with GLS inhibitors depletes both glutamate and glutathione in IDH mutant glioma models. Consistent with these findings, the glutaminase inhibitor CB-839 specifically sensitized IDH mutant glioma cells to oxidative stress in vitro and to radiation in vitro and in vivo [151]. CB-839 is now being evaluated in combination with radiation therapy and temozolomide in open-label Phase Ib clinical trials for patients with IDH-mutated diffuse or anaplastic astrocytoma (NCT03528642).

Inhibitors of nucleotide metabolism have been effectively combined with radiation across numerous malignancies for decades. This strategy has been explored in GBM as well. Gemcitabine, which inhibits deoxynucleotide formation by inhibiting ribonucleotide reductase, has been combined with radiation in the clinic, where it was well tolerated and yielded favorable outcomes in the pre-temozolomide era [179]. Whether inhibitors of nucleotide metabolism would have a favorable safety and efficacy profile when combined with temozolomide is not certain.

### 5.2. Chemotherapy

Alkylating chemotherapy is a standard of care in the treatment of GBM. Many patients with GBM exhibit intrinsic resistance to chemotherapy and those that do initially respond eventually develop recurrences as well [180,181]. Abnormal metabolic pathway utilization may contribute to this resistance phenotype and its inhibition may help overcome resistance to alkylating chemotherapy in GBM. Inhibition of glycolysis at the hexokinase step either with 3-bromopyruvate, 2-deoxyglucose or genetic knockdown of hexokinase confers modest sensitization to temozolomide in GBM cell lines and intracranial tumor models [61,182,183]. Dichloroacetate, which inhibits glycolysis more distally and promotes the oxidation of glucose-derived carbons in the mitochondria, combines less favorably temozolomide [184]. The mechanisms governing these favorable and unfavorable combinations have not been studied and it remains unclear what glucose-derived metabolites or co-factors are responsible for mediating temozolomide resistance. 

Non-glycolytic metabolic enzymes may also mediate temozolomide resistance. Long noncoding RNAs TP73-AS1 enhances the resistance of GBM cancer stem cells to TMZ by promoting the expression of ALDH1A1, which encodes the aldehyde dehydrogenase 1 family member A1 protein and is enriched in cancer stem cell populations. ALDH1 inhibitors, which mark cells with stem-like properties, increased the sensitivity of GBM cells to TMZ [185]. ALDH inhibitors are now being investigated in clinical trials in combination with temozlomide and radiation and preliminary results suggest that the combination is well tolerated and may have efficacy in subsets of GBM patients [186].

The nitrosourea class of chemotherapies is the main cytotoxic alternative to TMZ for patients with GBM [187]. 3-bromo-2-oxopropionate-1-propyl ester (3-BrOP), a more stable derivative of HK2 inhibitor 3-BrPA, depletes ATP by inhibiting glycolysis [188]. Glioblastoma stem cells (GSCs) that are resistant to single agent TMZ or the nitrosurea carmustine are sensitized to both by 3-BrOP, which depletes ATP and inhibits carmustine-induced DNA repair [189]. In addition to this intrinsic resistance, GSCs become resistant to nitrosureas when they are in hypoxic conditions. This hypoxia-induced nitrosurea resistance is also overcome when cells are treated by 3-BrPA analogs [190]. 

### 5.3. Targeted Therapy

Molecularly targeted therapies have shown little activity to date in GBM. Most GBMs exhibit activation of the PI3K/AKT/mTOR pathway, which provided a strong rationale for the use of mTOR inhibitors in this disease [11]. While some trials are still ongoing, initial reports of single agent mTOR inhibitors were disappointing [191,192]. Combining inhibition of mTOR with standard chemoradiation yielded similarly disappointing results [193]. Metabolic adaptation may underlie resistance to mTOR inhibition in GBM. Indeed, treatment with mTOR inhibitors increases GLS expression and glutamate levels and inhibition of GLS sensitizes GBM cell lines and xenografts to mTOR inhibitors [85]. Similar synergism between GLS and mTOR inhibitors is seen in other cancers as well [194], which suggests this combination may be worthy of clinical investigation in GBM as well as other diseases. 

Anti-angiogenic therapy has had minimal success in improving outcomes for patients with GBM [195,196]. Metabolic adaptation may also cause resistance to this family of targeted therapies. Treatment with bevacizumab increases the uptake of glucose and its conversion into lactate in orthotopic patient-derived xenograft models of GBM [197]. Similarly, bevacizumab treatment reduces the flow of glucose-derived carbons into the TCA cycle. Whether this metabolic adaption is a direct effect of bevacizumab or simply a result of bevacizumab-induced hypoxia is unknown [198]. Regardless, these observations suggest that the efficacy of bevacizumab could be augmented by a PDH activator such as DCA, but to our knowledge this combination has not been tested.

### 5.4. Immunotherapy

Immune-based therapies have become standard treatment options for many cancers [199,200]. These therapies are also under investigation in GBM, but results thus far have been disappointing. Inhibitors of immune checkpoint receptors have been used and are well tolerated in GBM but results to date have been mixed. Indeed, the CheckMate-143 clinical trial randomized 369 patients with recurrent GBM to either bevacizumab or nivolumab (an inhibitor of the programmed cell death 1 checkpoint) treatment. The one-year overall survival was approximately 40% in both arms and there were no changes in the median overall survival [201]. Similar results were seen in primary GBM in the CheckMate-498 trial, which randomized over 500 patients with *MGMT*-umethylated primary GBMs to treatment with either radiation with temozolomide or radiation with nivolumab. There was no significant improvement in survival in the nivolumab treatment arm [202]. Whether nivolumab will have efficacy in *MGMT*-methylated GBM will be determined in the pending CheckMate-548 trial. Despite, these disappointing results, a recent study showed that blockade of PD-1 signaling prior to surgical resection in recurrent GBM may be more efficacious than PD-1 administration after resection [203]. Thus, it is possible that the neoadjuvant immune checkpoint blockade could prove efficacious in both de novo and recurrent GBM. Other immune-based therapies including additional checkpoint inhibitors, chimeric antigen receptor (CAR) T cells, and viral-based therapies remain under investigation in GBM [204].

The metabolic environment of GBM may be immunosuppressive and limit the efficacy of immune-based therapies. Tumor-derived lactate suppresses T cell function by blocking lactate export and suppressing their ability to maintain aerobic glycolysis [205]. Lactate similarly inhibits type 1 interferon signaling [206]. Hence, the high lactate concentrations found within many GBMs may contribute to the disappointing clinical trial results of immunotherapies in this disease. Targeting lactate metabolism by either inhibiting GBM glycolysis of lactate export via the monocarboxylate transporter could thus be unexplored strategies to improve immunotherapy efficacy in this disease [207,208].

Most GBMs also express the metabolic enzyme indoleamine 2,3 dioxygenase 1 (IDO1), which catalyzes the conversion of tryptophan to kynurenine [209]. The metabolic consequences of IDO1 activity (e.g., low tryptophan and high kynurenine) suppress the activity of T cells through a variety of mechanisms and may also limit the efficacy of immunotherapy in GBM. In support of this hypothesis, pharmacologic inhibition of IDO augments the efficacy of immune checkpoint blockade in intracranial mouse models of GBM [210]. Unfortunately, initial clinical trial results combining IDO inhibitors with a checkpoint blockade in pancreatic cancer and melanoma were disappointing and enthusiasm for continued testing of this combination in clinical trials is waning [211]. 

## 6. Conclusions and Future 

The number of treatments that have impacted survival in GBM can be counted on a single hand. The past decade has seen tremendous elucidation of the genetic abnormalities and liabilities that characterize GBM, but these have not yet translated into useful clinical strategies. Targeting altered metabolism rather than genetic abnormalities is a promising alternative strategy that we hope may provide some benefit in this challenging disease. Inhibiting mutated IDH or exploiting the liabilities conferred by its metabolic product (R)-2HG represent promising metabolic strategies to improve outcomes in IDHmtGBM. Since the IDH mutation is an early oncogenic event, its presence (and the presence of (R)-2HG is relatively homogeneous in these tumors, which are otherwise largely characterized by heterogeneity. The path forward for metabolic therapy in IDHwtGBM is less clear. TMZ and radiation, two of the successful therapies in IDHwtGBM place intense metabolic demands on cells and we are hopeful that combining metabolic inhibitors with these treatments is perhaps the most reasonable first strategy to pursue.

## Figures and Tables

**Figure 1 cancers-11-01231-f001:**
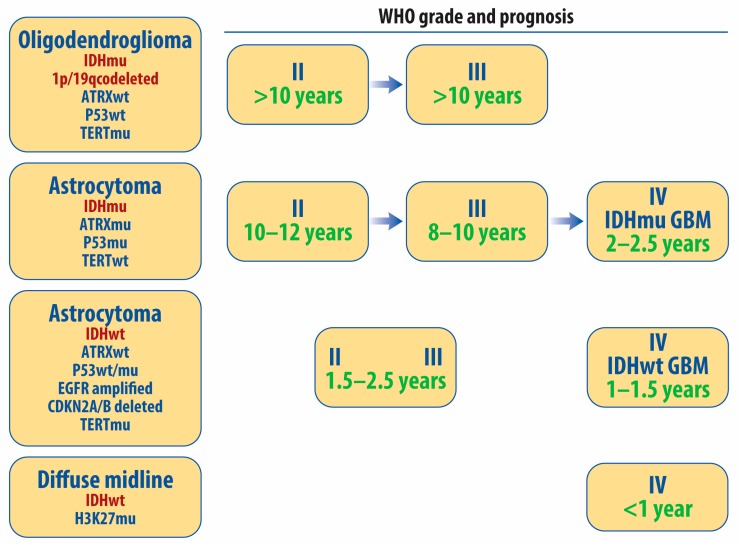
Classification of diffuse gliomas. As of 2016, diffuse gliomas are now defined based on both their molecular and histologic features. Definitional molecular features are noted in red, while other common molecular features are also listed. For most diffuse gliomas, the grade is still determined by the presence of conventionally defined “aggressive” microscopic features such as atypia, mitoses, microvascular proliferation, and necrosis. H3K27 mutant midline gliomas are grade 4 based on their molecular features. Median survival estimates for each type of molecularly defined tumor receiving optimal therapy are listed in green, though many of these estimates should be viewed as preliminary due to the recent reclassification of these tumors [4,33,34,35,36,37,38,39,40,41,42].

**Figure 2 cancers-11-01231-f002:**
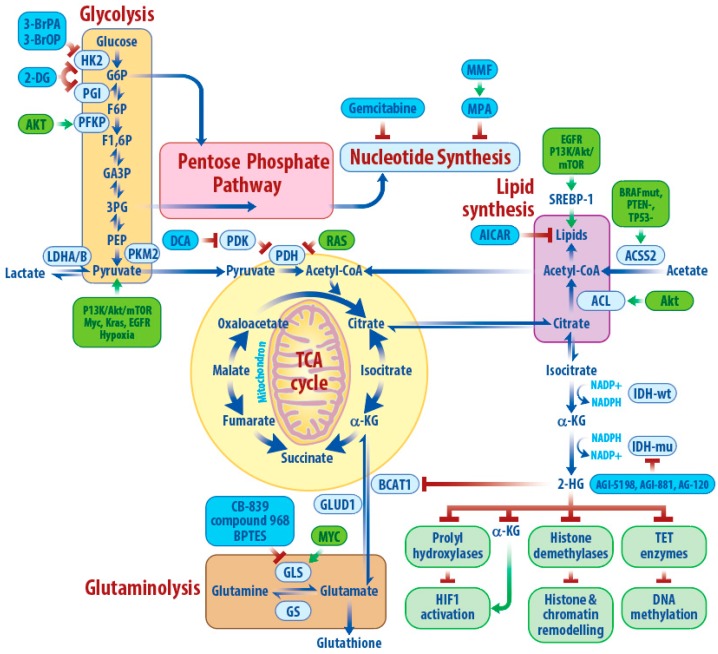
Key metabolic remodeling and potential therapeutic targets in glioblastoma (GBM). Metabolic reprogramming involved in GBM includes glycolysis, TCA cycle, glutaminolysis, pentose phosphate pathway (PPP), lipid and nucleotide synthesis, and D-2HG pathways. Key enzymes (light blue box) of these pathways can be regulated by known oncogenic signaling pathways (green box) and be targeted (dark blue box) for the treatment of GBM. Note: Blue arrows indicate metabolic conversion; dark red arrows indicate inhibition and green arrows indicate stimulation or activation.

**Figure 3 cancers-11-01231-f003:**
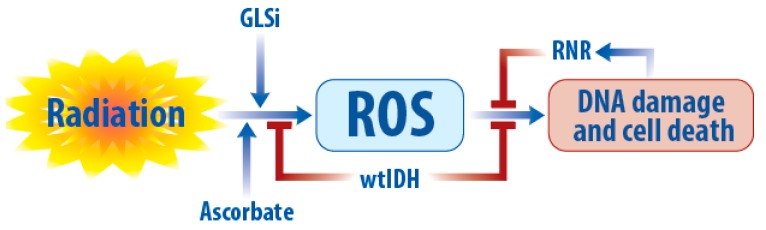
Summary of metabolic targeting in radio-sensitizing of GBM. Radio-sensitivity can be improved by depletion of glutathione (GLSi) and NAD+ (ascorbate and PARPi), and inhibition of IDH (wtIDHi or mIDHi) and deoxyribonucleotide (RNRi) to finally induce DNA damage and cell death.

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
