# Peer review of "Metabolic Abnormalities in Glioblastoma and Metabolic Strategies to Overcome Treatment Resistance"

_cancers, 2019, doi:10.3390/cancers11091231_

Round 1

Reviewer 1 Report

The review nicely lays out the reasons for the need to target metabolic pathways in glioblastoma and proposes strategies to do so in IDH mut gliomas.

Major concerns:

1-mixed oligoastrocytoma is no longer a distinct classification in WHO 2016. 1p19q codeletion status is now used to further identify these tumors. Please refer to the following figure from WHO 2016 classification, Louis et al, introduction, page 16. Figure 1 needs to be modified.

2-Line 171. It would beneficial to insert a short paragraph summarizing what metabolites that contribute to TCA cycle in GBMs.

3-For each category, it would help the reader to know about available clinical trials if the audience are MDs. For example, a trial of gluctaminase inhibitor in addition to PARP inhibitor are ongoing and accept GBM patients: https://clinicaltrials.gov/ct2/show/NCT03875313

4- section 3.5 requires a description regarding how salvage pathway and de novo pathway are different

5-  a description is needed to elaborate on potential reasons for increased sensitivity of IDH mut tumors to combination of TMZ and PARP inhibitors.

6- line 390, how can NAD+ be depleted?

7- line 414- if 2HG enhances the ability of glioma to repair DNA, then IDH mut tumors would be less susceptible to radiation. This paragraph needs to be rephrased.

8- line 522- any strategies to lower lactate levels to augment immunotherapy effects?

Minor concerns

Line 193, space between IDH and reference 85 is missing

Line 196, space between vivo and reference 77 is missing

Line 279, space between published and reference 105 is missing

Lines 323, metabolically switches

Line 408, change “whether it is efficacious” to “its efficacy in GMB”

Line 401, IDH1 is the most upregulated product?

Line 422- space between NAD+ and reference 150 is missing

Line 444- “inhibt” misspelled 

Line 511- need a reference after results of CheckMate -143

Line 514- space between arm and reference 187 is missing

Line 517- space between GBM and reference 188 is missing

Line 518- and limits

Line 523- space between kynurenine and ref 191 is missing

Line 528- space between GBM and ref 192 is missing

Line 530- space between waning and ref 193 is missing

Line 538- extra “(“?

Line 541 – IDHwtGBM, TMZ and radiation, place

Author Response

Dear Reviewers:

Thank you very much for your positive comments on our manuscript and your thoughtful critiques.  Below, we are addressing point-by-point your constructive critiques, and we have modified the text accordingly.

Review 1#

Major concerns:

“1-mixed oligoastrocytoma is no longer a distinct classification in WHO 2016. 1p19q codeletion status is now used to further identify these tumors. Please refer to the following figure from WHO 2016 classification, Louis et al, introduction, page 16. Figure 1 needs to be modified.”

Reply: We agree that figure 1 is outdated. We have updated Figure 1 and the accompanying text in the manuscript (section 2) and the Figure legend to reflect the molecular definitions of glioma in WHO 2016.

“2-Line 171. It would beneficial to insert a short paragraph summarizing what metabolites that contribute to TCA cycle in GBMs.”

Reply: We agree with the reviewer’s comments. We have reedited the TCA cycle section and made it clear in the introduction that acetate and glucose are thought to dominate carbon supply to the TCA cycle in GBM while glutamine appears to play a smaller role. (Line166-175 in the new version)

“3-For each category, it would help the reader to know about available clinical trials if the audience are MDs. For example, a trial of gluctaminase inhibitor in addition to PARP inhibitor are ongoing and accept GBM patients: https://clinicaltrials.gov/ct2/show/NCT03875313”

Reply: We have added the “ClinicalTrials.gov Identifier” number for each ongoing clinical trial including the one noted by the reviewer above, inhibitors of FASN (NCT03032484) and glutaminase inhibitors in combination with radiation (NCT03528642) and others.

“4- section 3.5 requires a description regarding how salvage pathway and de novo pathway are different”

Reply: We have better clarified the differences between nucleotide salvage and de novo synthesis (Line 313-316 in the new version).

“5-  a description is needed to elaborate on potential reasons for increased sensitivity of IDH mut tumors to combination of TMZ and PARP inhibitors.”

Reply: We have added a brief discussion of potential reasons why IDH mut tumors may have increased sensitivity to the combination of TMZ and PARP inhibitors. We believe the most plausible explanation is impaired homologous recombination due to R-2HG-mediated inhibition of the histone lysine demethylases.  (PMID 28148839, Line 415-420)

“6- line 390, how can NAD+ be depleted?”

Reply: We have clarified that 2HG inhibits one pathway of NAD+ generation by inhibiting Naprt1, which renders IDHmutant cells and tumors exquisitely sensitive to inhibition of NAMPT. (Line 434-440)

“7- line 414- if 2HG enhances the ability of glioma to repair DNA, then IDH mut tumors would be less susceptible to radiation. This paragraph needs to be rephrased.”

Reply: There is controversy regarding this topic. As noted by reviewer number 3, several groups have shown that IDH mutant tumors have impaired DNA repair and enhanced sensitivity to DNA damaging agents such as temozolomide, radiation and PARP inhibitors due to the ability of 2HG to suppress HR. Another group has recently published (PMID 30760578) that 2HG transcriptionally upregulates ATM and increases the activity of HR. We have added a more detailed discussion about this controversy (line 417-430).   

“8- line 522- any strategies to lower lactate levels to augment immunotherapy effects?”

 Reply: We have suggested several strategies to lower lactate levels in GBM, but to date none of these have been investigated in combination with immunotherapy in the laboratory or clinic (Line 581-584).

Minor concerns

“Line 193, space between IDH and reference 85 is missing”

Reply: We fixed this.

“Line 196, space between vivo and reference 77 is missing”

Reply: We added the space.

“Line 279, space between published and reference 105 is missing”

Reply: We added the space.

“Lines 323, metabolically switches”

Reply: We corrected this.

“Line 408, change “whether it is efficacious” to “its efficacy in GMB””

Reply: We changed it.

“Line 401, IDH1 is the most upregulated product?”

Reply: We corrected it. The updated sentence is “IDH1 is the most upregulated NADPH-producing enzyme in IDHwtGBM” (Line 458).

“Line 422- space between NAD+ and reference 150 is missing”

Reply: We added the space

“Line 444- “inhibt” misspelled”

Reply: We corrected it

“Line 511- need a reference after results of CheckMate -143”

Reply: Per reviewer’s suggestion, we added the reference (PMID: 2920768) in the revised version.

“Line 514- space between arm and reference 187 is missing”

Reply: We added the space.

“Line 517- space between GBM and reference 188 is missing”

Reply: We added it.

“Line 518- and limits”

Reply: We corrected it.

“Line 523- space between kynurenine and ref 191 is missing”

Reply: We added the space.

“Line 528- space between GBM and ref 192 is missing”

Reply: We added the space.

“Line 530- space between waning and ref 193 is missing”

Reply: We added the space.

“Line 538- extra “(“?”

Reply: We deleted it.

“Line 541 – IDHwtGBM, TMZ and radiation, place”

Reply: We reedited it.

We believe that we have addressed the critiques raised by the reviewers and hope you will find that it is now suitable for publication in Cancers.

Thank you very much for your consideration, and I look forward to hearing from you soon.

Best regards,

Daniel R. Wahl,

Assistant Professor of Radiation Oncology

University of Michigan Medical School

Reviewer 2 Report

This study reports the metabolic characteristics of glioblastoma (GBM) that remodel tumor metabolism to promote survival and proliferation. The authors discuss strategies to overcome the radio and chemotherapy resistance of GBM.

This study is exciting and summarizes the recent critical findings on the metabolic liabilities in GBM. The authors suggest that immunotherapy is not efficient against GBM. However, a recent study (Cloughesy TF et al., Nature Medicine 2019, PMID: 30742122) indicates that the inhibition of PD-1 enhances both the local and systemic antitumor immune response and may lead to a more effective approach to the treatment of GBM. The authors should discuss and cite this study.

Figure 2 is confusing and needs to be improved. For example, Nucleotide synthesis and pentose phosphate pathway (PPP) should not be included in the same box. MPA is an inhibitor of nucleotide synthesis but not of the PPP. Acetyl-CoA is drawn inside the TCA cycle. However, this metabolite is not a TCA cycle intermediate but a substrate of this pathway.

“Glycolysis” needs to be straightened.

LDHA/B, both isoforms, should be indicated.

2-DG is a more specific inhibitor of Hexokinase than 3-bromopyruvate, which blocks HK2 but also compete with pyruvate for LDH reaction.

Author Response

Dear Reviewers:

Thank you very much for your positive comments on our manuscript and your thoughtful critiques.  Below, we are addressing point-by-point your constructive critiques, and we have modified the text accordingly.

Reviewer 2#

“This study reports the metabolic characteristics of glioblastoma (GBM) that remodel tumor metabolism to promote survival and proliferation. The authors discuss strategies to overcome the radio and chemotherapy resistance of GBM.

This study is exciting and summarizes the recent critical findings on the metabolic liabilities in GBM. ”

Reply: We are very grateful to the reviewer for his/her positive and encouraging comments on our manuscript.

“The authors suggest that immunotherapy is not efficient against GBM. However, a recent study (Cloughesy TF et al., Nature Medicine 2019, PMID: 30742122) indicates that the inhibition of PD-1 enhances both the local and systemic antitumor immune response and may lead to a more effective approach to the treatment of GBM. The authors should discuss and cite this study.”

Reply: We have added this important study and emphasized the observation that moving checkpoint blockade to the neoadjuvant setting may provide benefit not seen when such agents are given after surgery (Line 571-576).

“Figure 2 is confusing and needs to be improved. For example, Nucleotide synthesis and pentose phosphate pathway (PPP) should not be included in the same box. MPA is an inhibitor of nucleotide synthesis but not of the PPP. Acetyl-CoA is drawn inside the TCA cycle. However, this metabolite is not a TCA cycle intermediate but a substrate of this pathway.

“Glycolysis” needs to be straightened.

LDHA/B, both isoforms, should be indicated.

2-DG is a more specific inhibitor of Hexokinase than 3-bromopyruvate, which blocks HK2 but also compete with pyruvate for LDH reaction”

Reply: We have extensively modified Figure 2 and believe the updated figure addresses each of these points.

We believe that we have addressed the critiques raised by the reviewers and hope you will find that it is now suitable for publication in Cancers.

Thank you very much for your consideration, and I look forward to hearing from you soon.

Best regards,

Daniel R. Wahl,

Assistant Professor of Radiation Oncology

University of Michigan Medical School

Reviewer 3 Report

Title: Metabolic abnormalities in glioblastoma and metabolic strategies to overcome treatment resistance

Authors: Zhou W and Wahl DR

Summary: This review article is timely, thorough, and well written; the authors should be commended for their work.  This review provides a welcome synthesis of the literature surrounding an emergent theme in brain tumor therapy: the development of new strategies that target tumor-specific metabolic alterations to overcome heterogeneity-driven treatment resistance.  Overall, I feel that the authors comprehensively address the dominant metabolic pathways that have been shown to be altered in gliomas and highlight interesting avenues to exploit these alterations therapeutically.  I have a few points that the authors may consider addressing but otherwise fully support publication of this manuscript. 

Conceptual points:

1)    The 2016 WHO classification guidelines for glioma de-emphasized the category of ‘mixed’ glioma through the introduction of molecular markers for diagnosis, which helps distinguish astrocytomas and oligodendrogliomas with ‘mixed’ histology.  I would consider revising Figure 1 to reflect this.

2)    Figure 1 could be edited to better represent the association of IDH mutations with each glioma classification.  Furthermore, differences in the natural histories of primary vs secondary GBM could be depicted more clearly. 

3)    Lines 75-79: To my knowledge TERT promoter mutations are mutually exclusive with ATRX LOF mutations.  If the authors have evidence to the contrary, it would be helpful to cite that here.  Also, expansion of this section to include a few additional studies may be helpful, as this is an important concept and may help link genetic alterations driving tumor progression with metabolic changes highlighted later in the manuscript.  In terms of discussing relevant mechanisms of glioma progression, I would consider mentioning the pathways and mutations highlighted in the following studies: PMIDs 29016839, 24714777, 26824661.

4)    There are a few sections where the authors might address certain nuances and conflicting findings in different studies that they highlight, including:

a.     Lines 108-118: Although multiple papers that the authors cite support a role for PKM2 as a protein kinase, there is also evidence to the contrary (PMID 26300261). 

b.     Lines 337-338: Past studies of mutant IDH1 enzymatic activity revealed dependence on WT IDH1 but a recent study showed that this dependence does not involve channeling between the monomers (PMID 30381394).  This may be worth mentioning here to paint a more complete picture.

c.     Lines 345-353: 2HG has been reported to increase (PMID 19359588) as well as decrease (PMID 22343896) levels of HIF1a in glioma-relevant cell culture models.  However, HIF target genes were shown to be expressed at lower levels in IDH mutant vs WT primary glioma patient samples in the latter study, suggesting that the dominant effect of IDH mutations in human tumors is repression rather than activation of HIF1a.  Acknowledgement of this complexity is warranted and could be represented more clearly in Figure 2.

d.     Lines 410-419: The issue of whether IDH mutations promote or inhibit DNA damage responses is one of the most important questions in the field now and will be crucial to determining how best to deploy mutant IDH inhibitors in combinatorial therapeutic regimens in the clinic.  The strong evidence (PMIDs 28148839, 26363012, 28202508) that mutant IDH inhibits the DNA damage response should at least be given equal weight in this section and the conclusions drawn should be more nuanced in my opinion.  Also, I believe [131] (Xu et al) is cited in error here.  Do they authors intend to refer to PMID 30760578 instead?

Minor Points:

1)    Line 40: spelling

2)    Lines 44-45: grammar

3)    Figure 2: BCAT1/2 promote, not inhibit, glutamate synthesis in glial cells

4)    Line 153: spelling

5)    Lines 299-304: conveying that nucleoside analogues generally display poor BBB penetrance is an important point here.

6)    Lines 322-324: grammar

7)    Line 334: don’t italicize references to IDH1 and IDH2 enzymes

8)    Line 365: data is plural, therefore use ‘suggest’ vs ‘suggests’

9)    Lines 384-391: consider citing PMID 27430238 here

10) Line 435: rational, not rationale

11) Line 473: spelling

12) Lines 476-484: spelling is nitrosourea, not nitrosurea

13) Lines 480-484: run-on sentence

14) Line 540: I would spell out the ‘successful therapies’ that are being referred to here for clarity

Author Response

Dear Reviewers:

Thank you very much for your positive comments on our manuscript and your thoughtful critiques.  Below, we are addressing point-by-point your constructive critiques, and we have modified the text accordingly.

Reviewer 3#

“Summary: This review article is timely, thorough, and well written; the authors should be commended for their work.  This review provides a welcome synthesis of the literature surrounding an emergent theme in brain tumor therapy: the development of new strategies that target tumor-specific metabolic alterations to overcome heterogeneity-driven treatment resistance.  Overall, I feel that the authors comprehensively address the dominant metabolic pathways that have been shown to be altered in gliomas and highlight interesting avenues to exploit these alterations therapeutically.  I have a few points that the authors may consider addressing but otherwise fully support publication of this manuscript.”

Reply: We really appreciate the review’s compliments and encouragements on our manuscript.

Conceptual points:

“1)    The 2016 WHO classification guidelines for glioma de-emphasized the category of ‘mixed’ glioma through the introduction of molecular markers for diagnosis, which helps distinguish astrocytomas and oligodendrogliomas with ‘mixed’ histology.  I would consider revising Figure 1 to reflect this.”

“2) Figure 1 could be edited to better represent the association of IDH mutations with each glioma classification.  Furthermore, differences in the natural histories of primary vs secondary GBM could be depicted more clearly.”

Reply: We have extensively revised Figure 1 and the text in the manuscript regarding the WHO 2016 classification of glioma and the natural histories of primary and secondary GBMs.

“3)    Lines 75-79: To my knowledge TERT promoter mutations are mutually exclusive with ATRX LOF mutations.  If the authors have evidence to the contrary, it would be helpful to cite that here.  Also, expansion of this section to include a few additional studies may be helpful, as this is an important concept and may help link genetic alterations driving tumor progression with metabolic changes highlighted later in the manuscript.  In terms of discussing relevant mechanisms of glioma progression, I would consider mentioning the pathways and mutations highlighted in the following studies: PMIDs 29016839, 24714777, 26824661.”

Reply: We agree with the reviewer that TERT promoter mutations and ATRX LOF mutations are usually mutually exclusive, though they can co-occur in rare occasions (26061753). We have also extensively edited section 2 to provide an overview of glioma progression in light of the WHO 2016 re-classificaiton (Line 54-88 in the new version).

“4)    There are a few sections where the authors might address certain nuances and conflicting findings in different studies that they highlight, including:”

“a.     Lines 108-118: Although multiple papers that the authors cite support a role for PKM2 as a protein kinase, there is also evidence to the contrary (PMID 26300261).” 

Reply: We have added a brief description of this controversy at line 135-137.

“b.     Lines 337-338: Past studies of mutant IDH1 enzymatic activity revealed dependence on WT IDH1 but a recent study showed that this dependence does not involve channeling between the monomers (PMID 30381394).  This may be worth mentioning here to paint a more complete picture.”

Reply: We have added this reference to our manuscript and noted that it implies an absence of substrate channeling (Line 366-370).

“c.     Lines 345-353: 2HG has been reported to increase (PMID 19359588) as well as decrease (PMID 22343896) levels of HIF1a in glioma-relevant cell culture models.  However, HIF target genes were shown to be expressed at lower levels in IDH mutant vs WT primary glioma patient samples in the latter study, suggesting that the dominant effect of IDH mutations in human tumors is repression rather than activation of HIF1a.  Acknowledgement of this complexity is warranted and could be represented more clearly in Figure 2.”

Reply: We addressed this complexity in both the text and Figure 2.

“d.     Lines 410-419: The issue of whether IDH mutations promote or inhibit DNA damage responses is one of the most important questions in the field now and will be crucial to determining how best to deploy mutant IDH inhibitors in combinatorial therapeutic regimens in the clinic.  The strong evidence (PMIDs 28148839, 26363012, 28202508) that mutant IDH inhibits the DNA damage response should at least be given equal weight in this section and the conclusions drawn should be more nuanced in my opinion.  Also, I believe [131] (Xu et al) is cited in error here.  Do they authors intend to refer to PMID 30760578 instead?”

Reply: We added a paragraph describing this controversy and updated the reference in question (Line 416-428).

Minor Points:

“1)    Line 40: spelling”

Reply: We corrected the word “genomic”.

“2)    Lines 44-45: grammar”

Reply: We corrected it.

“3)    Figure 2: BCAT1/2 promote, not inhibit, glutamate synthesis in glial cells”

Reply: The reviewer is correct. We are sorry for this mistake. We corrected it in Figure 2.

“4)    Line 153: spelling”

Reply: We corrected it.

“5)    Lines 299-304: conveying that nucleoside analogues generally display poor BBB penetrance is an important point here.”

Reply: We agree. Although previous studies reported that nucleoside analogues (like Gemcitabine) poorly penetrated the BBB, later studies demonstrated that the uptake of gemcitabine in brain tumors is much higher than the tumor-free regions and gemcitabine shows favorable efficacy when combined with radiation in gliomas patients. We discussed these points in our revised version (Line 329-331).

“6)    Lines 322-324: grammar”

Reply: We reedited this paragraph.

“7)    Line 334: don’t italicize references to IDH1 and IDH2 enzymes”

Reply: We corrected it.

“8)    Line 365: data is plural, therefore use ‘suggest’ vs ‘suggests’”

Reply: We corrected it.

“9)    Lines 384-391: consider citing PMID 27430238 here”

Reply: Thanks for the reviewer’s good suggestion. We cited the reference (Line 430).

“10) Line 435: rational, not rationale”

Reply: We corrected it.

11) Line 473: spelling

Reply: We corrected it.

12) Lines 476-484: spelling is nitrosourea, not nitrosurea

Reply: We changed “nitrosurea” to “nitrosourea” in this paragraph.

13) Lines 480-484: run-on sentence

Reply: We modified it (527-529).

14) Line 540: I would spell out the ‘successful therapies’ that are being referred to here for clarity

Reply: We spelled out the two therapies, “TMZ and radiation”, in our revised manuscript.

We believe that we have addressed the critiques raised by the reviewers and hope you will find that it is now suitable for publication in Cancers.

Thank you very much for your consideration, and I look forward to hearing from you soon.

Best regards,

Daniel R. Wahl,

Assistant Professor of Radiation Oncology

University of Michigan Medical School

Reviewer 4 Report

This review article is exhaustive in volume but lacks depth and novelty. This review touch base the whole literature published with regards to GBM and its targets with a little emphasis towards understanding the tumor metabolism. A lot many avenues were mentioned but was not discussed clearly. 

Author Response

Dear Reviewers:

Thank you very much for your positive comments on our manuscript and your thoughtful critiques.  Below, we are addressing point-by-point your constructive critiques, and we have modified the text accordingly.

Reviewer 4#

“This review article is exhaustive in volume but lacks depth and novelty. This review touch base the whole literature published with regards to GBM and its targets with a little emphasis towards understanding the tumor metabolism. A lot many avenues were mentioned but was not discussed clearly.” 

Reply: We designed this review article to provide an overview of the metabolic abnormalities characteristic of GBM and then more deeply describe how metabolism mediates therapy resistance (section 5). We have extensively clarified the language and discussion of each of the sections and hope that this manuscript will serve as a reference for members of the scientific community who are interested in a broad overview of GBM metabolism or designing new combinatorial therapies.

We believe that we have addressed the critiques raised by the reviewers and hope you will find that it is now suitable for publication in Cancers.

Thank you very much for your consideration, and I look forward to hearing from you soon.

Best regards,

Daniel R. Wahl,

Assistant Professor of Radiation Oncology

University of Michigan Medical School

Round 2

Reviewer 4 Report

The authors revised the review as suggested and is ready for publication.